# Language models can learn complex molecular distributions

Daniel Flam-Shepherd ⬤ [1,2 ✉], Kevin Zhu[1] & Alán Aspuru-Guzik ⬤ [1,2,3,4 ✉]

Deep generative models of molecules have grown immensely in popularity, trained on relevant datasets, these models are used to search through chemical space. The downstream utility of generative models for the inverse design of novel functional compounds, depends on their ability to learn a training distribution of molecules. The most simple example is a language model that takes the form of a recurrent neural network and generates molecules using a string representation. Since their initial use, subsequent work has shown that language models are very capable, in particular, recent research has demonstrated their utility in the low data regime. In this work, we investigate the capacity of simple language models to learn more complex distributions of molecules. For this purpose, we introduce several challenging generative modeling tasks by compiling larger, more complex distributions of molecules and we evaluate the ability of language models on each task. The results demonstrate that language models are powerful generative models, capable of adeptly learning complex molecular distributions. Language models can accurately generate: distributions of the highest scoring penalized LogP molecules in ZINC15, multi-modal molecular distributions as well as the largest molecules in PubChem. The results highlight the limitations of some of the most popular and recent graph generative models– many of which cannot scale to these molecular distributions.

[1] Department of Computer Science, University of Toronto, Toronto, ON M5S 2E4, Canada. [2] Vector Institute for Artificial Intelligence, Toronto, ON M5S 1M1, Canada. [3] Department of Chemistry, University of Toronto, Toronto, ON M5G 1Z8, Canada. [4] Canadian Institute for Advanced Research, Toronto, ON M5G 1Z8, Canada. ✉email: danielfs@cs.toronto.edu; alan@aspuru.com

The efficient exploration of chemical space is one of the most important objectives in all of science, with numerous applications in therapeutics and materials discovery. However, exploration efforts have only probed a very small subset of the synthetically accessible chemical space[1], therefore developing new tools is essential. The rise of artificial intelligence may provide the methods to unlock the mysteries of the chemical universe, given its success in other challenging scientific questions like protein structure prediction[2].

Very recently, deep generative models have emerged as one of the most promising tools for this immense challenge[3]. These models are trained on relevant subsets of chemical space and can generate novel molecules similar to their training data. Their ability to learn the training distribution and generate valid, similar molecules—is important for success in downstream applications like the inverse design of functional compounds.

The first models involved re-purposing recurrent neural networks (RNNs)[4] to generate molecules as SMILES strings[5]. These language models can be used to generate molecular libraries for drug discovery[6] or built into variational autoencoders (VAE)[3,7] where bayesian optimization can be used to search through the model's latent space for drug-like molecules. Other models generate molecules as graphs either sequentially[8–14] using graph neural networks[15,16] or generate whole molecules in one shot[17–20]. Two of the most popular: CGAVE and JTVAE can be directly constrained to enforce valency restrictions. Other models generate molecules as point clouds in 3D space[21].

Language models have been widely applied[22] with researchers using them for ligand-based de novo design[23]. A few recent uses of language models include: targeting natural-product-inspired retinoid X receptor modulators[24], designing liver X receptor agonists[25], generating hit-like molecules from gene expression signatures[26], designing drug analogs from fragments[27], composing virtual quasi-biogenic compound libraries[28] and many others. Additional studies have highlighted the ability of language models

in the low-data regime[29,30] with improved performance using data augmentation[31].

Initially the brittleness of the SMILES string representation meant a single character could lead to invalid molecules. This problem has been largely solved with more robust molecular string representations[32–35]. Additionally, with improved training methods, deep generative models based on RNNs consistently generate a high proportion of valid molecules using SMILES[6,9,36]. One area that has not been studied is the ability of language models and generative models to generate larger more complex molecules or generate from chemical spaces with large ranges in size and structure. This is beneficial because of increased interest in larger more complex molecules for therapeutics[37].

To test the ability of language models, we formulate a series of challenging generative modeling tasks by constructing training sets of more complex molecules than exist in standard datasets[3,36,38]. In particular, We focus on the ability of language models to learn the distributional properties of the target datasets. We train language models on all tasks and baseline many other graph generative model as well—although we focus on CGAVE and JTVAE. The results demonstrate that language models are powerful generative models and can learn complex molecular distributions better than most graph generative models.

## Results

We define three tasks, generating: (1) distributions of molecules with high scores of penalized LogP[3] (Fig. 1a, d), (2) multi-modal distributions of molecules (Fig. 1b, e), and (3) the largest molecules in PubChem (Fig. 1c, e). Necessarily, each different generative modeling task is defined by learning to generate from the distribution of molecules in a dataset. We build three datasets using relevant subsets of larger databases.

In Table 1 there are some summary statistics of atom and ring number in all datasets compared with two standard datasets Zinc[3]

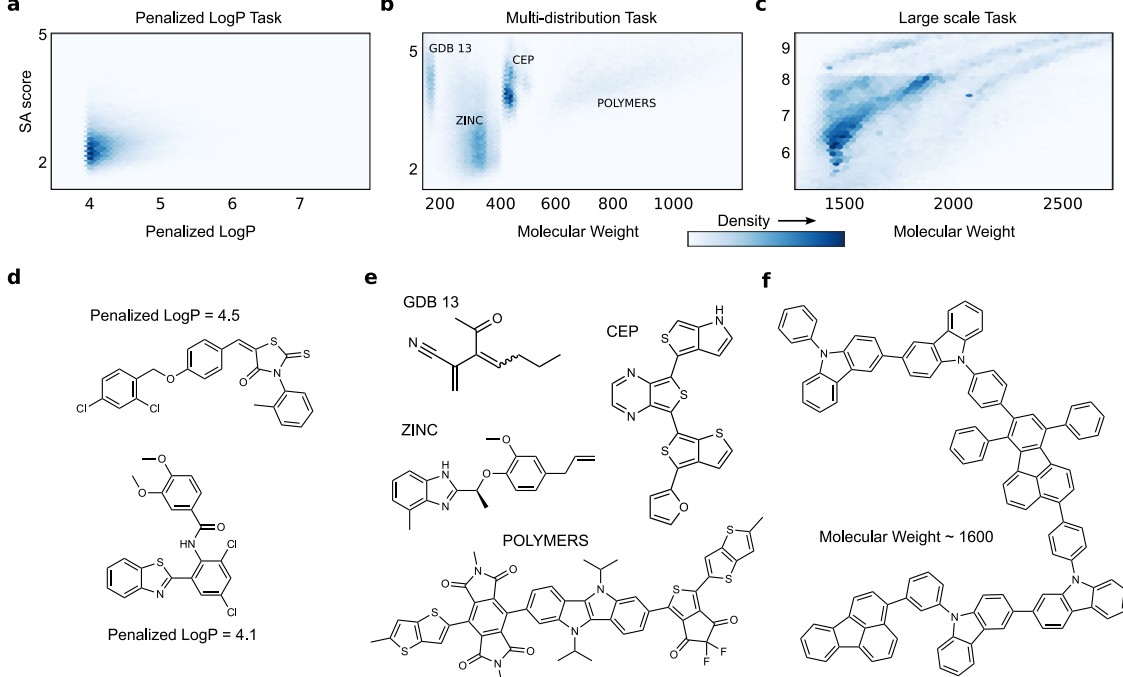

**Fig. 1 The generative modeling tasks. a–c** The molecular distributions defining the three complex molecular generative modeling task. **a** The distribution of penalized LogP vs. SA score from the training data in the penalized logP task. **b** The four modes of differently weighted molecules in the training data of the multi-distribution task. **c** Large scale task's molecular weight training distribution. **d–f** examples of molecules from the training data in each of the generative modeling tasks. **d** The penalized LogP task, **e** The multi-distribution task. **f** The large-scale task.

and Moses[36]. All tasks involve larger molecules with more sub-structures and contain a larger range of atom and ring number per molecule.

For each task we assess performance by plotting the distribution of training molecules properties and the distribution learned by the language models and graph models. We use a histogram for the training molecules and fit a Gaussian kernel density estimator to it by tuning its bandwidth parameter. We plot KDE's for molecular properties from all models using the same bandwidth parameter.

From all models we initially generate 10K (thousand) molecules, compute their properties and use them to produce all plots and metrics. Furthermore, for fair comparison of learned distributions, we use the same number of generated molecules from all models after removing duplicates and training molecules.

For quantitative evaluation of any model's ability to learn its training distribution, we compute the Wasserstein distance between property values of generated molecules and training molecules. We also compute the Wasserstein distance between different samples of training molecules in order to determine a most optimal baseline, which we can compare with as an oracle.

**Table 1 Dataset statistics for all three tasks compared to standard datasets.**

|  | # Atoms | | | # Rings | | |
|---|---|---|---|---|---|---|
|  | Min | Mean | Max | Min | Mean | Max |
| Zinc | 6 | 23.2 | 38 | 0 | 2.8 | 9 |
| Moses | 8 | 21.6 | 27 | 0 | 2.6 | 8 |
| LogP | 12 | 34.7 | 78 | 0 | 4.2 | 37 |
| Multi | 7 | 31.1 | 106 | 0 | 5.3 | 23 |
| Large | 101 | 140.1 | 891 | 0 | 11.2 | 399 |

For molecular properties we consider: quantitative estimate of drug-likeness (QED)[39], synthetic accessibility score (SA)[40], octanol–water partition coefficient (Log $P$)[41], exact molecular weight (MW), Bertz complexity (BCT)[42], natural product likeness (NP)[22]. We also use standard metrics like validity, uniqueness, novelty– to assess the model's ability to generate a diverse set of real molecules distinct from the training data.

For models, our main consideration is a chemical language model using a recurrent neural network with long short-term memory[43] and is trained on SMILES (SM-RNN) or SELFIES (SF-RNN). We also train two of the most popular deep graph generative models: the junction tree variational autoencoder (JTVAE)[10] and the constrained graph variational autoencoder (CGVAE)[9].

**Penalized LogP Task**. For the first task, we consider one of the most widely used benchmark assessments for searching chemical space, the penalized LogP task—finding molecules with high LogP[44] penalized by synthesizability[40] and unrealistic rings. We consider a generative modeling version of this task, where the goal is to learn distributions of molecules with high penalized LogP scores. Finding individual molecules with good scores (above 3.0) is a standard challenge but learning to directly generate from this part of chemical space, so that every molecule produced by the model has high penalized LogP, adds another degree of difficulty. For this we build a training dataset by screening the ZINC15 database[45] for molecules with values of penalized LogP exceeding 4.0. Many machine learning approaches can only find a handful of molecules in this range, for example JTVAE[10] found 22 total during all their attempts. After screening, the top scoring molecules in ZINC amounted to roughly 160K (K is thousand) molecules for the training data in this task. Thus, the training distribution is extremely spiked with most density falling around 4.0–4.5 penalized LogP as seen in Fig. 1a with most training molecules resembling the examples

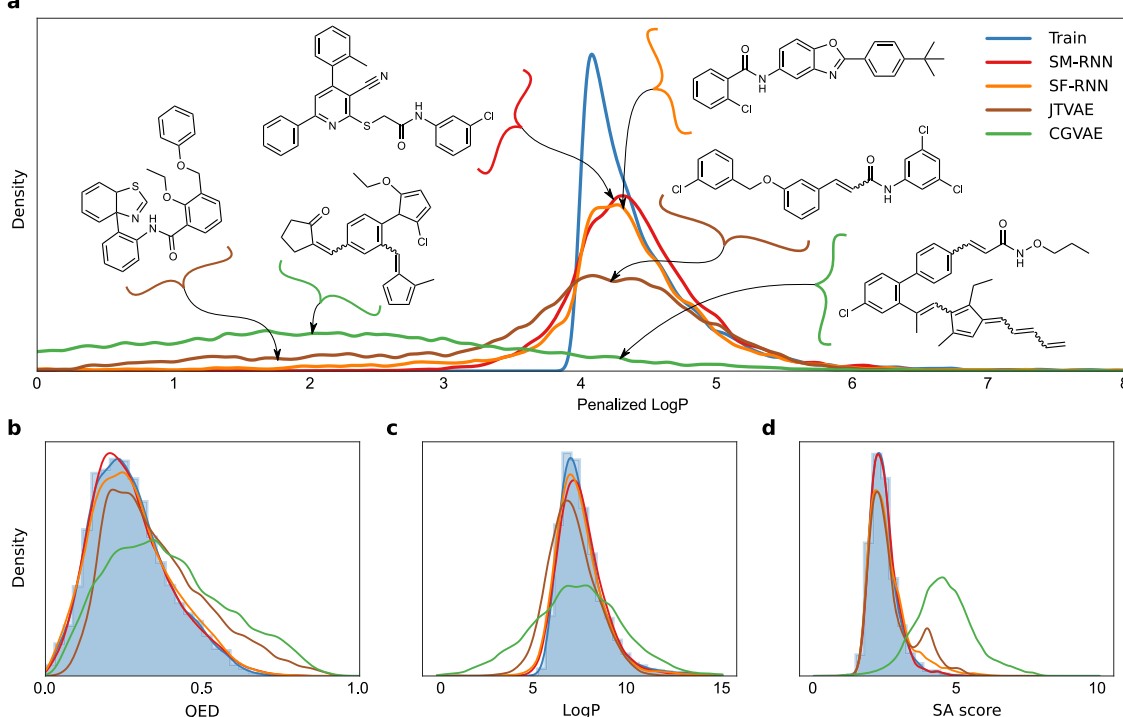

**Fig. 2 Penalized LogP Task I. a** The plotted distribution of the penalized LogP scores of molecules from the training data (TRAIN) with the SM-RNN trained on SMILES, the SF-RNN trained on SELFIES and graph models: CGVAE and JTVAE. For the graph models we display molecules from the out of distribution mode at penalized LogP score ∈ [1.75, 2.25] as well as molecules with penalized LogP score in the the main mode [4.0,4.5] from all models. **b**–**d** Distribution plots for all models and training data of molecular properties QED, LogP, and SA score.

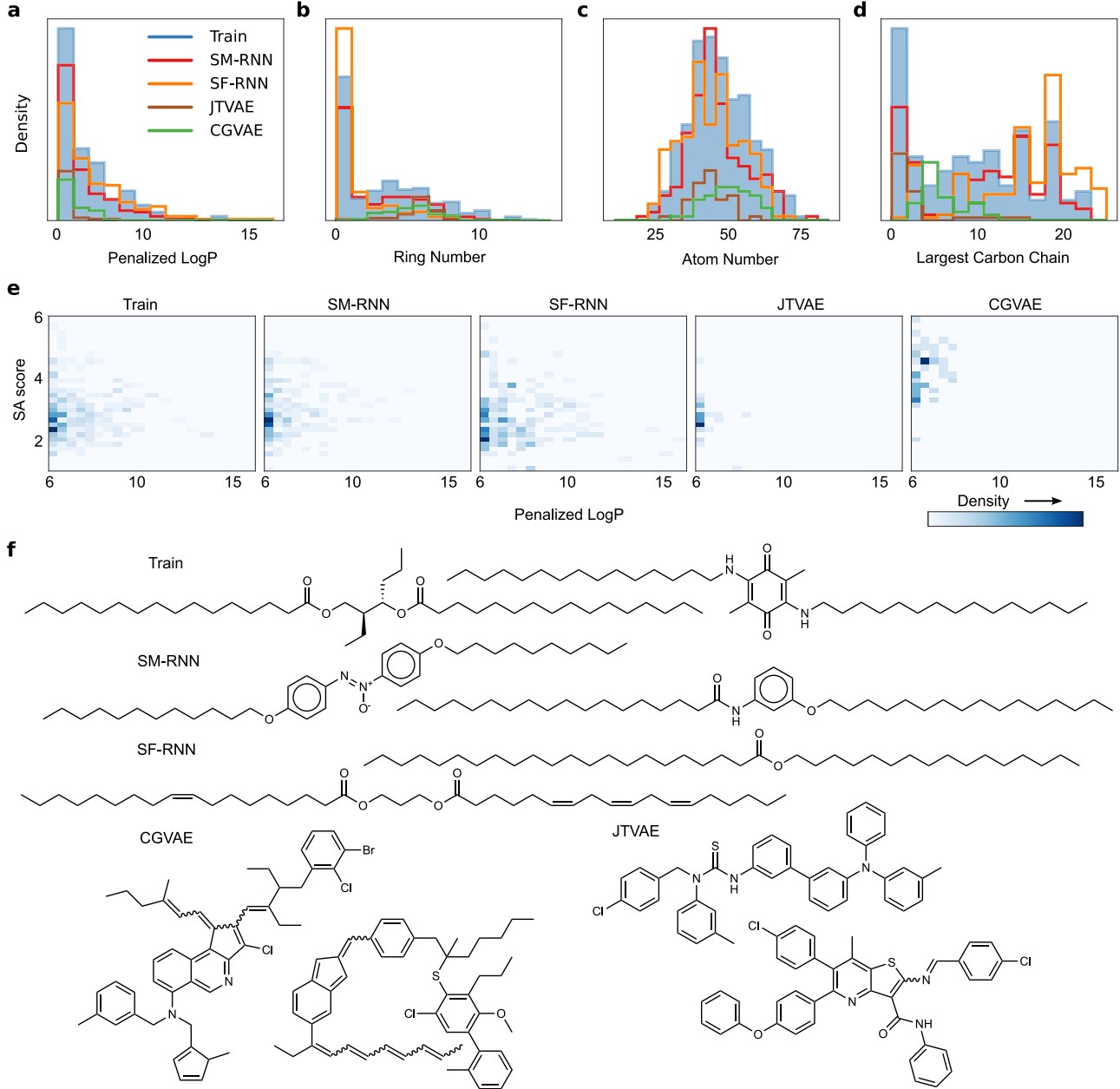

**Fig. 3 Penalized LogP Task II. a–d** Histograms of penalized LogP, Atoms #, Ring # and length of largest carbon chain (all per molecule) from molecules generated by all models or from the training data that have penalized LogP ≥ 6.0. **e** 2d histograms of penalized LogP and SA score from molecules generated by the models or from training data that have penalized LogP ≥ 6.0. **f** A few molecules generated by all models or from the training data that have penalized LogP ≥ 6.0.

shown in Fig. 1d. However, some of the training molecules, around 10% have even higher penalized LogP scores—adding a subtle tail to the distribution.

The results of training all models are shown in Figs. 2 and 3. The language models perform better than the graph models, with the SELFIES RNN producing a slightly closer match to the training distribution in Fig. 2a. The CGVAE and JTVAE learn to produce a large number of molecules with penalized LogP scores that are substantially worse than the lowest training scores. It is important to note, from the examples of these shown in Fig. 2a these lower scoring molecules are quite similar to the molecules from the main mode of the training distribution, this highlights the difficulty of learning this distribution. In Fig. 2b–d we see that

JTVAE and CGVAE learn to produce more molecules with larger SA scores than the training data, as well, we see that all models learn the main mode of LogP in the training data but the RNNs produce closer distributions– similar results can be seen for QED. These results carryover for quantitative metrics and both RNNs achieve lower Wasserstein distance metrics than the CGVAE and JTVAE (Table 2) with the SMILES RNN coming closest to the TRAIN oracle.

We further investigate the highest penalized LogP region of the training data with values exceeding 6.0—the subtle tail of the training distribution. In the 2d distributions (Fig. 3e) it's clear that both RNNs learn this subtle aspect of the training data while the graph models ignore it almost completely and only learn

**Table 2 Wasserstein distance metrics for LogP, SA, QED, MW, BT, and NP between molecules from the training data and generated by the models for all three tasks.**

| Task | Samples | LogP | SA | QED | MW | BCT | NP |
|---|---|---|---|---|---|---|---|
| LogP | TRAIN | 0.020 | 0.0096 | 0.0029 | 1.620 | 7.828 | 0.013 |
| | SM-RNN | 0.095 | 0.0312 | 0.0068 | 3.314 | 21.12 | 0.054 |
| | SF-RNN | 0.177 | 0.2903 | 0.0095 | 6.260 | 25.00 | 0.209 |
| | JTVAE | 0.536 | 0.2886 | 0.0811 | 35.93 | 76.81 | 0.164 |
| | CGVAE | 1.000 | 2.1201 | 0.1147 | 69.26 | 141.2 | 1.965 |
| Multi | TRAIN | 0.048 | 0.0158 | 0.0020 | 2.177 | 14.149 | 0.010 |
| | SM-RNN | 0.081 | 0.0246 | 0.0059 | 5.483 | 21.118 | 0.012 |
| | SF-RNN | 0.286 | 0.1791 | 0.0227 | 11.35 | 68.809 | 0.079 |
| | JTVAE | 0.495 | 0.2737 | 0.0343 | 27.71 | 171.87 | 0.109 |
| | CGVAE | 1.617 | 1.8019 | 0.0764 | 30.31 | 183.58 | 1.376 |
| Large | TRAIN | 0.293 | 0.030 | 0.0003 | 18.92 | 85.04 | 0.005 |
| | SM-RNN | 1.367 | 0.213 | 0.0034 | 124.49 | 363.0 | 0.035 |
| | SF-RNN | 1.095 | 0.342 | 0.0099 | 67.322 | 457.5 | 0.111 |
| | JTVAE | – | – | – | – | – | – |
| | CGVAE | – | – | – | – | – | – |

TRAIN is an oracle baseline-values closer to it are better.

molecules that are closer to the main mode. In particular, CGVAE learns molecules with larger SA score than the training data. Furthermore, the molecules with highest penalized LogP scores in the training data typically contain very long carbon chains and fewer rings (Fig. 3b, d)—the RNNs are capable of picking up on this. This is very apparent in the samples the model produce, a few are show in Fig. 3f, the RNNs produce mostly molecules with long carbon chains while the CGVAE and JTVAE generate molecules with many rings that have penalized LogP scores near 6.0. The language models learn a distribution that is close to the training distribution in the histograms of Fig. 3a–d. Overall, the language models could learn distributions of molecules with high penalized LogP scores, better than the graph models.

**Multi-distribution task.** For the next task, we created a dataset by combining subsets of: (1) GDB13[46] molecules with molecular weight (MW) $\leq 185$, (2) ZINC[3,45] molecules with $185 \leq MW \leq 425$, (3) Harvard clean energy project (CEP)[47] molecules with $460 \leq MW \leq 600$, and the (4) POLYMERS[48] molecules with MW $> 600$. The training distribution has four modes– (Figs. 1b, e and 4a). CEP & GDB13 make up 1/3 and ZINC & POLYMERS take up 1/3 each of ~200K training molecules.

In the multi-distribution task, both RNN models capture the data distribution quite well and learn every mode in the training distribution (Fig. 4a). On the other hand, JTVAE entirely misses the first mode from GDB13 then poorly learns ZINC and CEP. As well, CGVAE learns GDB13 but underestimates ZINC and entirely misses the mode from CEP. More evidence that the RNN models learn the training distribution more closely is apparent in Fig. 4e where CGVAE and JTVAE barely distinguish the main modes. Additionally, the RNN models generate molecules better resembling the training data (Supplementary Table 4). Despite this, all models– except CGVAE, capture the training distribution of QED, SA score and Bertz Complexity (Fig. 4b–d). Lastly, in Table 2 the RNN trained on SMILES has the lowest Wasserstein metrics followed by the SELFIES RNN then JTVAE and CGVAE.

**Large-scale task.** The last generative modeling task, involves testing the ability of deep generative models to learn large molecules, the largest possible molecules relevant to molecular generative models that use SMILES/SELFIES string representations or

graphs. For this we turn to PubChem[49] and screen for the largest molecules with more than 100 heavy atoms, producing ~300K molecules. These are molecules of various kinds: small biomolecules, photovoltaics and others. They also have a wide range of molecular weight from 1250 to 5000 but most molecules fall into the 1250–2000 range (Fig. 1c).

This task was the most challenging for the graph models, both failed to train and were entirely incapable of learning the training data. In particular, JTVAE's tree decomposition algorithm applied to the training data produced a fixed vocabulary of ~11,000 substructures. However, both RNN models were able to learn to generate molecules as large and as varied as the training data. The training molecules correspond to very long SMILES and SELFIES string representations, in this case, the SELFIES strings provided an additional advantage and the SELFIES RNN could match the data distribution more closely (Fig. 5a). In particular, learning valid molecules is substantially more difficult with the SMILES grammar, as there are many more characters to generate for these molecules and a higher probability that the model will make a mistake and produce an invalid string. In contrast, the SELFIES string generated will never be invalid. Interestingly, even when the RNN models generated molecules that were out of distribution and substantially smaller than the training molecules —they still had similar substructures and resemblance to the training molecules (Fig. 5a). In addition, the training molecules seemed to be divided into two modes of molecules with lower and higher LogP values (Fig. 5b): with biomolecules defining the lower mode and molecules with more rings and longer carbons chains defining the higher LogP mode (more example molecules can be seen in supplementary Fig. 8). The RNN models were both able to learn the bi-modal nature of the training distribution.

The training data has a variety of different molecules and substructures, in Fig. 6a the RNN models adequately learn the distribution of substructures arising in the training molecules. Specifically the distribution for the number of: fragments, single atom fragments as well as single, fused-ring and amino acid fragments in each molecule. As the training molecules get larger and occur less, both RNN models still learn to generate these molecules (Fig. 5a when molecular weigh >3000).

The dataset in this task contains a number of peptides and cyclic peptides that arise in PubChem, we visually analyze the samples from the RNNs to see if they are capable of preserving backbone chain structure and natural amino acids. We find that

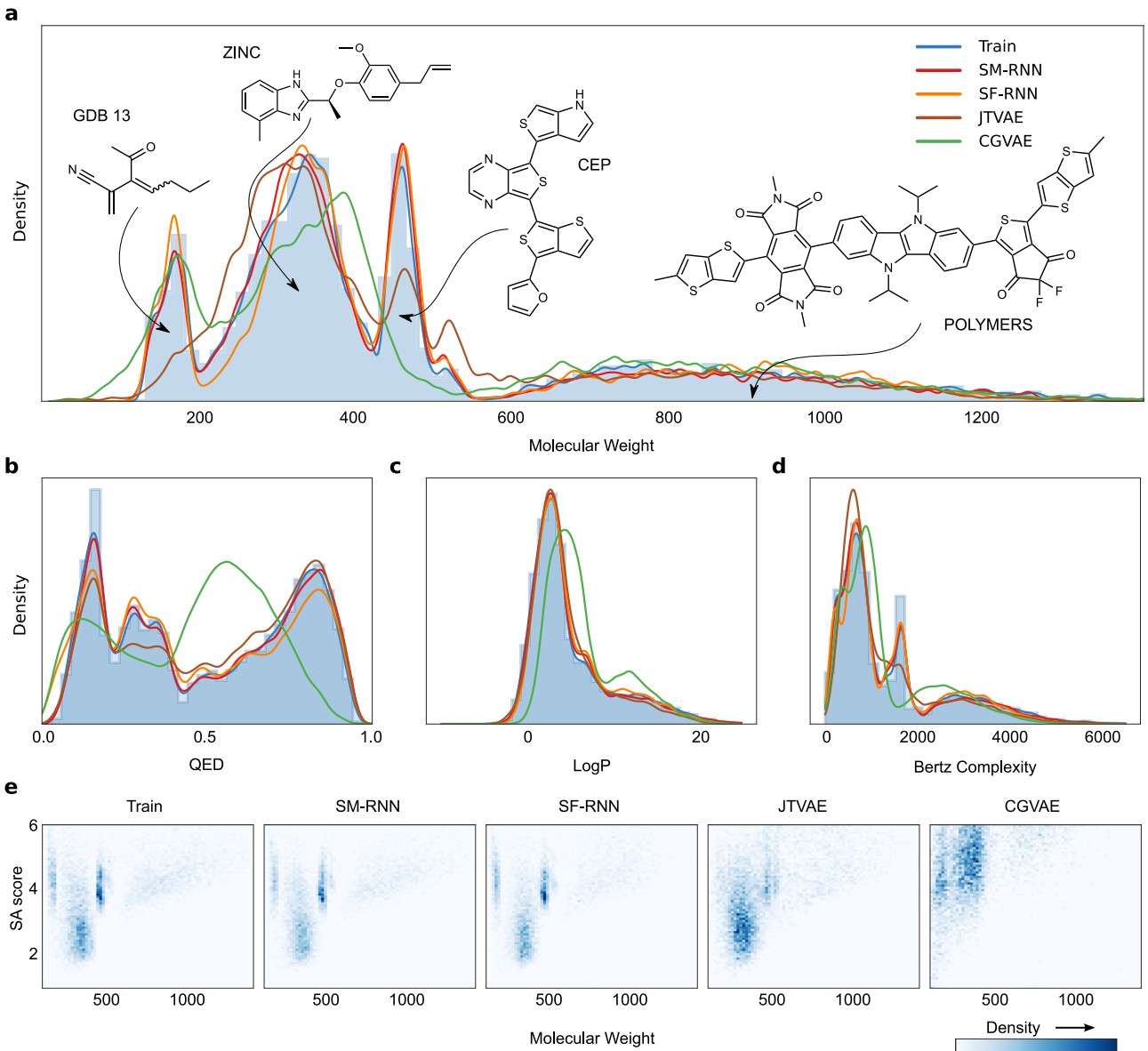

**Fig. 4 Multi-distribution Task. a** The histogram and KDE of molecular weight of training molecules along with KDEs of molecular weight of molecules generated from all models. Three training molecules from each mode are shown. **b–d** The histogram and KDE of QED, LogP and SA scores of training molecules along with KDES of molecules generated from all models. **e** 2d histograms of molecular weight and SA score of training molecules and molecules generated by all models.

the RNNs often sample snippets of backbone chains which are usually disjoint—broken up with other atoms, bonds and structures. In addition, usually these chains have standard side chains from the main amino acid residues but other atypical side chains do arise. In Fig. 6c we show two examples of peptides that are generated by the SM-RNN and SF-RNN. While there are many examples where both models do not preserve backbone and fantasize weird side-chains, it is very likely, that if trained entirely on relevant peptides the model could be used for peptide design. Even further, since these language models are not restricted to generating amino acid sequences that could be used to design any biochemical structure that mimic the structure of peptics or even replicate their biological behavior. This makes them very applicable to design modified peptides[50], other peptide mimetics and complex natural products[51,52]. The only requirement would be for a domain expert to construct a training dataset for specific targets. We conduct an additional study on how well the RNNs

learned the biomolecular structures in the training data, in Fig. 6b we see both RNNs match the distribution of essential amino acid (found using a substructure search). Lastly, it is also likely that the RNNs could also be used to design cyclic peptides. To highlight the promise of language models for this task we display molecules generated by the RNNs with the largest Tanimoto similarity to colistin and vancomycin (Fig. 6d). The results in this task demonstrate that language models could be used to design more complex biomolecules.

We also evaluate models on standard metrics in the literature: validity, uniqueness and novelty. Using the same 10K molecules generated from each model for each task we compute the following statistics defined in ref. [17] and store them in Table 3: (1) validity: the ratio between the number of valid and generated molecules, (2) uniqueness: the ratio between the number of unique molecules (that are not duplicates) and valid molecules, (3) novelty: the ratio between unique molecules that are not in the

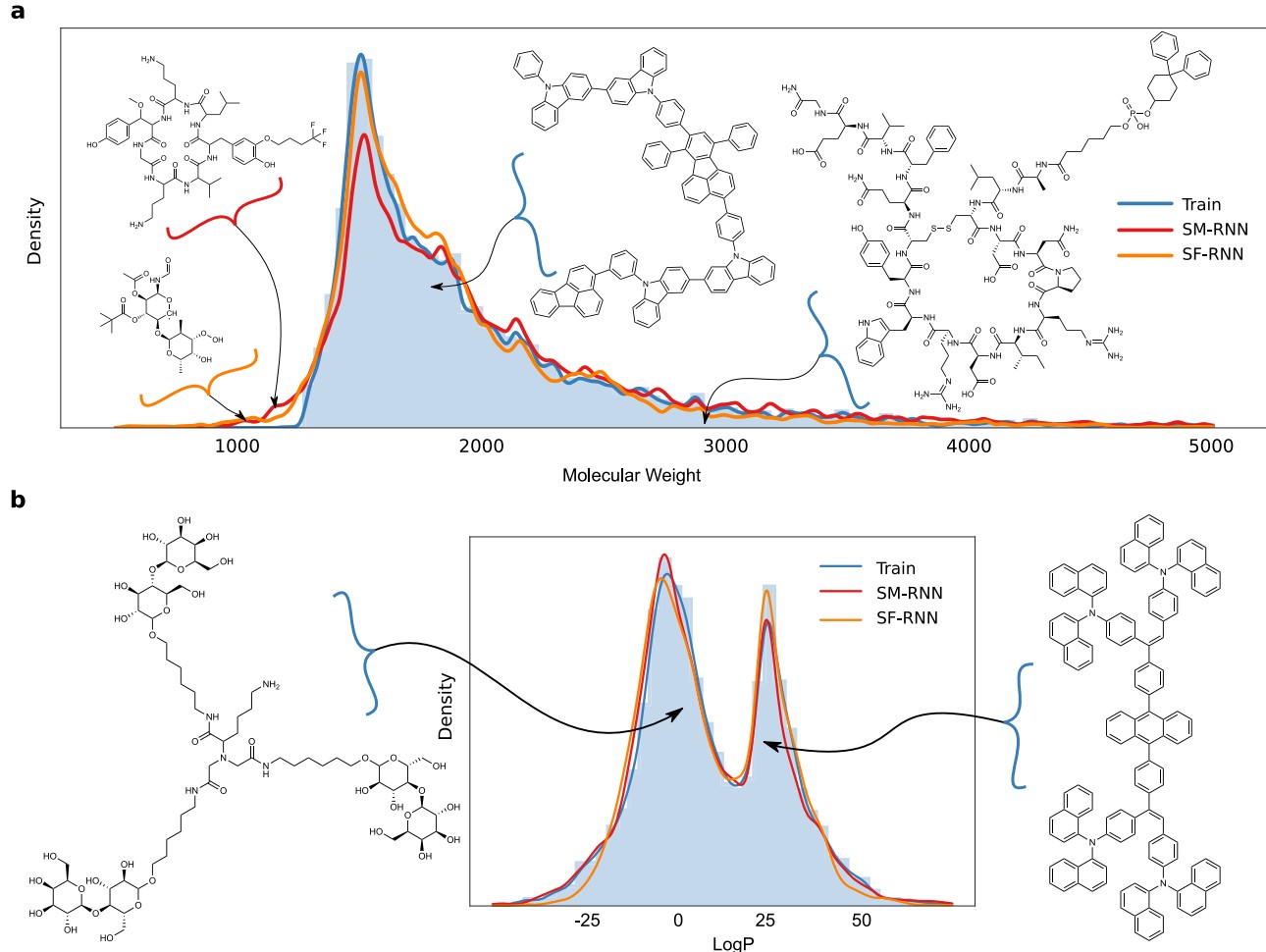

**Fig. 5 Large-scale Task I. a** The histogram and KDE of molecular weight of training molecules along with the KDEs of molecular weight of molecules generated from the RNNs. Two molecules generated by the RNN's with lower molecular weight than the training molecules are shown on the left of the plot. In addition, two training molecules from the mode and tail of the distribution of molecular weight are displayed on the right. **b** The histogram and KDE of LogP of training molecules along with the KDEs of LogP of molecules generated from the RNNs. On either side of the plot, for each mode in the LogP distribution, we display a molecule from the training data.

training data and the total number of unique molecules. In the first two tasks (Table 3), JTVAE and CGVAE have better metrics with very high validity, uniqueness and novelty (all close to 1), here the SMILES and SELFIES RNN perform worse but the SELFIES RNN is close to their performance. The SMILES RNN has the worse metrics due to its poor grammar but is not substantially worse than the other models.

We also considered many additional graph generative model baselines[8,12,17,19,53–58] on all tasks. These include some GANs[11,19], some autoregressive models[8,53,57], normalizing flows[54,58] and single shot models[17] Most do not scale at all and the few baselines that do—could only handle the LogP and multi-distribution tasks, but do not perform better than the language models. Results are shown in Supplementary Tables 1, 2 and Fig. 1.

## Discussion

In this work, in effort to test the ability of chemical language models, we introduce three complex modeling tasks for deep generative models of molecules. Language models and graph baselines perform each task, which entails learning to generate molecules from a challenging dataset.s The results demonstrate that language models are very powerful, flexible models that can learn a variety of very different complex distributions while the popular graph baselines are much less capable.

In comparison of SELFIES and SMILES, both the SM-RNN and SF-RNN perform well in all tasks, better than the baselines. We report that the SF-RNN has better standard metrics (Table 3) in every task, but the SM-RNN has better Wasserstein distance metrics (Table 2). Furthermore, the SF-RNN has better novelty than the SM-RNN—this may mean that the SELFIES grammar leads to less memorization of the training dataset in language models. This could also help explain why the SF-RNN has better standard metrics but worse Wasserstein metrics than the SM-RNN. In addition, data augmentation and random SMILES[31] could be used to improve the novelty score of the SM-RNN. In future, it would be valuable to have a more comprehensive evaluation of the use of SMILES and SELFIES representations in deep generative models.

The results show that the main baseline graph generative models, JTVAE and CGVAE are not as flexible as language models. For the penalized LogP task, the difference between a molecule that has a score of 2 and one that scores 4 often can be very subtle. Sometimes changing a single carbon or other atom can cause a large drop in score—this likely explains why the CGVAE severely misfit the main training mode. For the multi-distribution task, JTVAE and CGVAE's difficulties are clear but

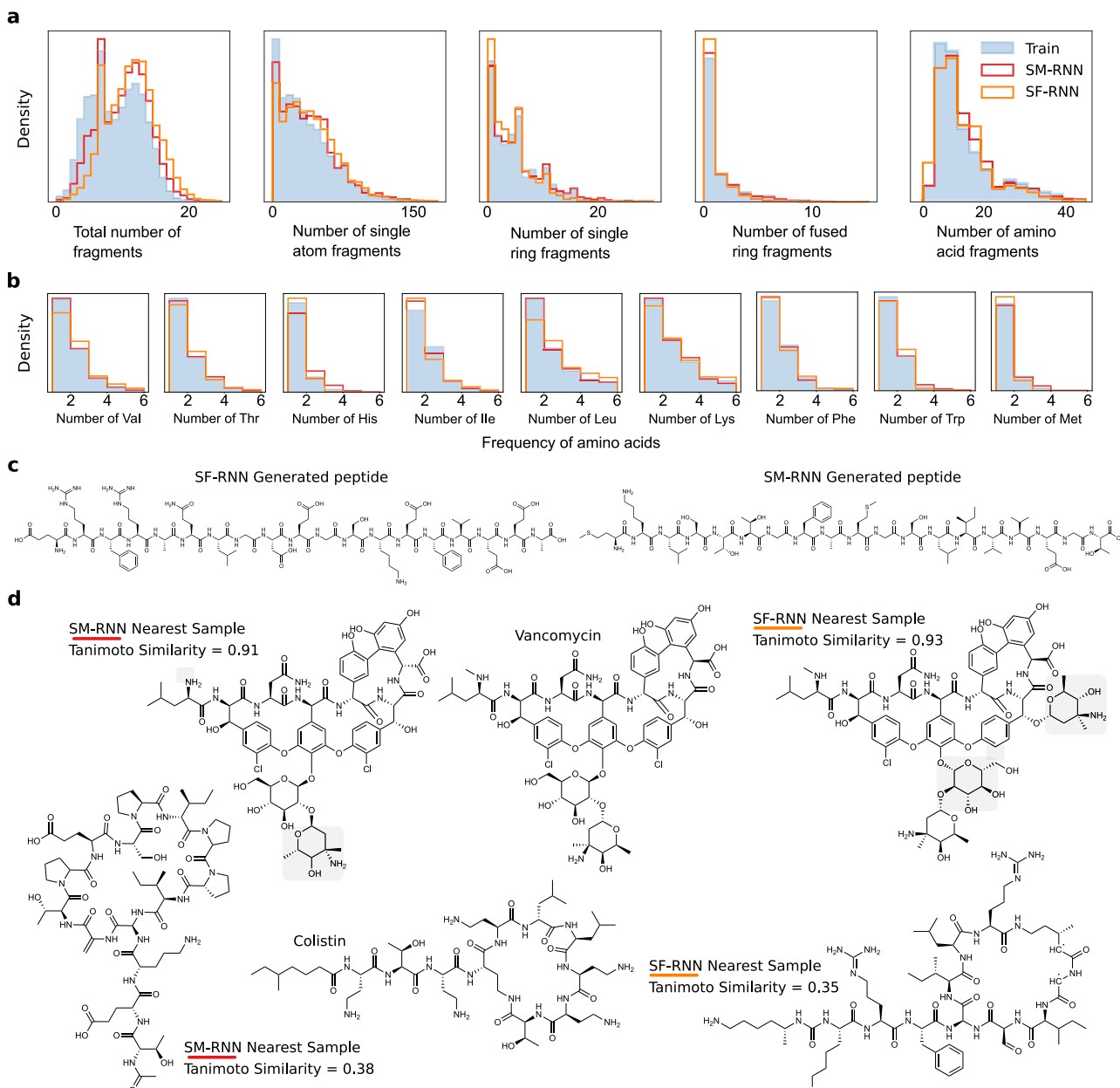

**Fig. 6 Large-scale Task II. a** Histograms of fragment #, single atom fragment #, single ring fragment #, fused-ring fragment #, amino acid fragment # (all per molecule) from molecules generated by the RNN models or from the training data. **b** Histograms of specific amino acid number in each molecule generated by the RNNs or from the training data. **c** A peptide generated by the SM-RNN—MKLSTTGFAMGSLIVVEGT (right) and one generated by the SF-RNN—ERFRAQLGDEGSKEFVEEA (left). **d** Molecules generated by the SF-RNN and SM-RNN that are closest in Tanimoto similarity to colistin and vancomycin. The light gray shaded regions highlight differences from vancomycin.

very understandable. For JTVAE, it has to learn a wide range of tree types: many of which have no large substructures like rings (the GDB13 molecules) while others are entirely rings (CEP and POLYMERS). For CGVAE, it has to learn a wide range of very different generation traces—which is difficult especially since it only uses one sample trace during learning. For the same reasons, these models were incapable of training on the largest molecules in PubChem.

The language models also perform better than the additional graph generative baselines—which have the same limitations as JTVAE and CGVAE. This is almost expected, as graph generative models have the more difficult task of generating both the atom and bond information—while a language model only has to generate a single sequence. Given this– it is natural that language

models display such flexible capacity and the evaluated graph generative models do not. Outside of molecular design some graph generative models have attempted to scale to larger graphs[59,60] but these models have not been augmented for molecules. The results here do highlight the fact that many widely used graph generative models are designed only for small drug-like[39] molecules and do not scale to larger more complex molecules. On the other hand, while language models can scale and flexibly generate larger molecules, graph generative models are more interpretable[53,57] which is important for drug and material discovery.

Based on the experiments conducted, language models are very powerful generative models for learning any complex molecular distribution and should see even more widespread use. However,

**Table 3 Standard metrics validity, uniqueness and novelty of molecules generated by all models in every task.**

| Task | Metric | SM-RNN | SF-RNN | JTVAE | CGVAE |
|------|--------|--------|--------|-------|-------|
| LogP | Validity | 0.941 | 1.000 | 1.000 | 1.000 |
| | Uniqueness | 0.987 | 1.000 | 0.982 | 1.000 |
| | Novelty | 0.721 | 0.871 | 0.980 | 1.000 |
| Multi | Valid | 0.969 | 1.000 | 0.999 | 0.999 |
| | Uniqueness | 0.996 | 0.989 | 0.998 | 0.996 |
| | Novelty | 0.937 | 0.950 | 0.998 | 1.000 |
| Large | Valid | 0.876 | 1.000 | – | – |
| | Uniqueness | 0.999 | 0.994 | – | – |
| | Novelty | 0.999 | 0.999 | – | – |

Closer to 1.0 indicates better performance.

it is still possible to see improvements to these models as these models cannot account for other important information like molecular geometry. In addition, we hope that the molecular modeling tasks and datasets introduced can motivate new generative models for these larger, more complex molecules. Future work will explore how capable chemical language models are in learning larger and larger snapshots of chemical space.

## Methods

**Hyper-parameter optimization**. For hyper-parameter optimization we use the simplest most effective method—namely random search[61]. We randomly sample from discrete grids of hyper-parameters with equal probability of selection for each value. The values are roughly equally spaced with 3–5 values in each grid. The upper and lower bounds for each hyper-parameter are defined as such: learning rate $\in [0.001, 0.0001]$, hidden units $\in [100, 1000]$, layer number $\in [1, 5]$, dropout (probability) in $[0.0, 0.5]$. We do not optimize the number of epochs– we just use the default value for the baseline models used during training on other datasets (MOSES, ZINC or Chembl).

**Model selection criteria**. There are many model selection criteria possible, for example—the MOSES benchmark[36] suggest the Frechet Distance, however, this and other performance metrics have been shown to have issues[62]. We evaluate and select models using all metrics employed in combination with the distribution plots. First we compile the top 10% of models with highest validity, uniqueness and novelty. Then we plot distribution plots for the main property of interest (i.e. penalized logP for LogP task and molecular weight for others)—then take the model that has the closest distribution to the training distribution and scores the lowest on largest number of the six Wasserstein distance metrics.

**Further details**. Language models are implemented in Python 3 with PyTorch[63] molecules are processed and relevant properties are computed using RDkit[64]. Wasserstein distances are computed using SciPy[65] as scipy.stats.wasserstein_-distance based on[66]—also known as the earth mover's distance, it can be viewed as the minimum amount of distribution weight that must be moved, multiplied by the distance—in order to transform samples from one distribution into samples from the another.

**Penalized LogP task details**. For the SM-RNN we used an LSTM with 2 hidden layer with 400 units and dropout in the last layer with prob = 0.2 and learning rate of 0.0001. For the SF-RNN we used an LSTM with 2 hidden layer with 600 units and dropout in the last layer with prob = 0.4 and learning rate of 0.0002. The CGVAE used 8 propagation layers and hidden layer side of 100 with kl annealed to 0.1 and a learning rate of 0.0015. The JTVAE used a learning rate of 0.001 and 3 GNN layers with a hidden size of 356.

**Multi-distribution task**. For the SM-RNN we used an LSTM with 3 hidden layer with 512 units and dropout in the last layer with prob = 0.5 and learning rate of 0.0001. For the SF-RNN we used an LSTM with 2 hidden layer with 500 units and dropout in the last layer with prob = 0.2 and learning rate of 0.0003. The CGVAE used 8 propagation layers and hidden layer side of 100 with kl annealed to 0.1 and a learning rate of 0.001. The JTVAE used a learning rate of 0.0001 and 3 GNN layers with a hidden size of 356.

**Large-scale task**. For the SM-RNN we used an LSTM network with 2 hidden layers with 512 units and dropout in the last layer with prob = 0.25 and learning

rate of 0.001. For the SF-RNN we used an LSTM network with 2 hidden layers with 800 units and dropout in the last layer with prob = 0.4 and learning rate of 0.0001.

## Data availability

The processed data used in this study are available in https://github.com/danielflamshep/genmoltasks.

## Code availability

The code used to train models is publicly available. JTVAE: https://github.com/wengong-jin/icml18-jtnn. CGVAE: https://github.com/microsoft/constrained-graph-variational-autoencoder. The RNN models were trained using the char-rnn code from https://github.com/molecularsets/moses. Trained models are available upon request.

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

## Acknowledgements

A.A.-G. acknowledge funding from Dr. Anders G. Frøseth. A.A.-G. also acknowledges support from the Canada 150 Research Chairs Program, the Canada Industrial Research Chair Program, and from Google, Inc. Models were trained using the Canada Computing Systems[67].

## Author contributions

D.F.-S. conceived the overall project, designed the experiments, prepared the datasets and wrote the paper. D.F.-S. and K.Z. trained the models and analyzed results. A.A.-G. led the project and provided overall directions.

## Competing interests

The authors declare no competing interests.
