## [Peer Review File · Nature Communications]

REVIEWER COMMENTS

Reviewer #1 (Remarks to the Author):

Aspuru-Guzik and coworkers theoretically analyze the performance of machine learning models for molecule design on challenging tasks and show superiority of language inspired models over graph-based approaches. The design tasks are well chosen to compare the models but their applicability on actual molecule design problems is at least partly elusive. The study provides useful insights for those working in the field especially by extending molecule design to very large molecules which hasn't been done with language models. The practical lessons learned from this study and their applicability to machine learning based design (apart from language based models being superior) should be better clarified. The manuscript is quite specialized but deserves publication after a few points have been addressed as outlined below.

- The problem of validity is exaggerated in the introduction. In fact, language models perform very well if properly trained and generate more than enough valid and chemically diverse molecules which has been shown in multiple studies. The study provides useful insights based on other metrics than validity.

- Page 1 line 32: the authors state that researches have attempted to use language models for de novo design. This is understated. There have been several practical applications of language models to design new molecules whose intended activity was experimentally confirmed. The reference cited here (8) is only one of many examples. It would be appropriate here to describe language models as validated by several application studies for de novo design and refer to recent progress in fine-tuning and intrinsic prioritization of designs.

- Language models have been applied to design small drug-like molecules based on SMILES and peptides based on single-letter amino acid code. The present study now extends to the design of larger molecules (including peptides) based on the SMILES and SELFIES representation. This is a strong and novel aspect of this study and it would add value to follow up with further studies. Some suggestions that - in the opinion of this reviewer - would make interesting analyses: Is a SMILES- or SELFIES-trained RNN able to conserve the peptide backbone, i.e., can the model design natural peptides? Does the model conserve natural amino acids? Can a SMILES based RNN be trained to specifically design cyclic or bi-/tri-/...cyclic peptides? Can such models design peptide mimetics? Such analysis would also benefit from direct comparisons between templates and designs, and an evaluation of training strategies.

Reviewer #2 (Remarks to the Author):

The manuscript by Flam-Shepherd et al. argues that generative models of molecules based on textual representations (aka language models) outperform generative models based on graphical representations (aka graph generative models). Their argument is based on a benchmark of two language models (both LSTMs, trained either on SELFIES or SMILES representations) against two graph generative models: constrained graph VAEs (Liu et al., NeurIPS 2018) and junction tree VAE (Jin et al., ICML 2018). The authors introduce three new tasks on which to benchmark these four models: specifically, generating molecules with high logP values, multi-modal property distributions, or with >100 heavy atoms. They demonstrate that the two language models are better at matching the target distributions on the first two tasks than either graph-based model.

I don't think the authors' results are necessarily novel in the strictest sense, because a handful of benchmarks have previously identified that language models outperform graph-based models (e.g., Mahmood et al., Nat. Commun. 2021). However, this has never really been a major focus of these analyses and I think it is potentially of enormous benefit for the field to highlight this more directly. With that being said, the major flaw in the paper is the level of evidence provided for the broad claim that language models outperform graph-based models. The authors have really only shown that language models outperform the two specific graph-based models evaluated here. Dozens of graph-based models have been described, as helpfully reviewed by Mercado et al., Applied AI Lett. 2020, and it is not clear why these specific two were selected (they are claimed to be "state of the art" but no specific evidence supporting this claim is provided). While I don't think it is either realistic or necessary to benchmark every single graph-based model that has ever been described, I don't think that the findings presented here convincingly support the more general claim. In my view, that would require benchmarking several (at least a half dozen) of the other prominent graph-based models for which source code is publicly available - for example GraphINVENT, MolGAN, GraphVAE, MolecularRNN, MolMP/MolRNN, HierVAE, and NeVAE - in order to establish a more general trend.

A second concern is the authors' argument that their own SELFIES representation "seems to improve the performance of language models in every task," as compared to a language model trained on SMILES. However, on 10 of 10 distribution learning tasks presented in Tables I and II, SM-RNN achieves better performance than SF-RNN. Thus, this statement is not supported and in fact is contradicted by the data presented in the manuscript.

A final issue is the level of detail provided in the Methods section, which is insufficient to reproduce all the results of the paper - for example, how was random search executed? How were the parameter grids defined? How were the evaluation metrics calculated? Ideally, the authors would just provide the source code that was used to execute these searches and evaluate model performance.

Minor points:

1. The statement in the abstract that graph generative models “typically achieve state of the art results” is not, to my knowledge, true, and in fact would seem to be contradicted by the results in this manuscript. It would be nice if the authors could clarify what data supports this claim or else remove it.
2. Much is made in the introduction of the notion that generation of invalid SMILES makes it difficult to train and apply language models. I can’t say that I understand why this is a major obstacle, as it would appear straightforward to filter out the invalid SMILES strings. I wonder if the authors can articulate some scenarios where it is essential that all generated molecules be valid (i.e. where filtering out invalid SMILES is not possible).
3. The authors assert that the three new benchmark tasks introduced here are “especially challenging.” I can see why modelling very large molecules could be challenging, but I’m not convinced that modelling especially lipophilic or chemically diverse molecules should present a particular challenge.
4. A table of Wasserstein metrics for the large scale task is missing.
5. In the Discussion, the authors suggest their results raise the possibility that language models are overfit to the training data. It was not clear to me exactly what aspects of their results suggested this. If overfitting is a concern, the framework based on the GDB-13 database presented by Arus-Pous et al., *J. Cheminform.* 2019 could be useful to test the generalization capacity of these models.

Dear Reviewers,

Thank you for your reviews. We have conducted a major revision of our paper based on your feedback.

We list the specific changes below :

- 1) We revise our introduction (**Response 1.1**): we add a more detailed discussion of relevant literature and further expand the discussion on the merits of our study.
- 2) We present additional analysis and discussion on the potential of language models to design peptides and cyclic peptides (**Response 1.3**).
- 3) We add additional benchmarks using several other prominent graph-based models (**Response 2.2**) and we expand the methods section with all implementation details (**Response 2.5**).
- 4) We add a more nuanced discussion on the abilities of graph generative models versus language models– avoiding broad claims (**Response 2.2**).

Reviewer 1 my response starts here and ends on page 4.

Reviewer 2 please see pages 5-12 for my response.

Dear Reviewer # 1,

Thank you for your suggestions, which have helped us strengthen our paper. Based on your review, we have revised our paper and conducted additional investigations. We go through each of your concerns as they arise throughout your review:

Reviewer #1 (Remarks to the Author):

Aspuru-Guzik and coworkers theoretically analyze the performance of machine learning models for molecule design on challenging tasks and show superiority of language inspired models over graph-based approaches. The design tasks are well chosen to compare the models but their applicability on actual molecule design problems is at least partly elusive. The study provides useful insights for those working in the field especially by extending molecule design to very large molecules which hasn't been done with language models. The practical lessons learned from this study and their applicability to machine learning based design (apart from language based models being superior) should be better clarified. The manuscript is quite specialized but deserves publication after a few points have been addressed as outlined below.

- The problem of validity is exaggerated in the introduction. In fact, language models perform very well if properly trained and generate more than enough valid and chemically diverse molecules which has been shown in multiple studies. The study provides useful insights based on other metrics than validity.

- Page 1 line 32: the authors state that researches have attempted to use language models for de novo design. This is understated. There have been several practical applications of language models to design new molecules whose intended activity was experimentally confirmed. The reference cited here (8) is only one of many examples. It would be appropriate here to describe language models as validated by several application studies for de novo design and refer to recent progress in fine-tuning and intrinsic prioritization of designs.

Response 1.1: We have updated the intro to have less emphasis on validity. We also expand on the motivation and merits of the study. We have added numerous additional references about language models being used for molecular design including :

1. Ertl, Peter, et al. "In silico generation of novel, drug-like chemical matter using the LSTM neural network." *arXiv preprint arXiv:1712.07449* (2017).
2. Awale, Mahendra, et al. "Drug analogs from fragment-based long short-term memory generative neural networks." *Journal of chemical information and modeling* 59.4 (2019): 1347-1356.
3. Méndez-Lucio, Oscar, et al. "De novo generation of hit-like molecules from gene expression signatures using artificial intelligence." *Nature communications* 11.1 (2020): 1-10.
4. Gupta, Anvita, et al. "Generative recurrent networks for de novo drug design." *Molecular informatics* 37.1-2 (2018): 1700111.
5. Blaschke, Thomas, et al. "Memory-assisted reinforcement learning for diverse molecular de novo design." *Journal of cheminformatics* 12.1 (2020): 1-17.
6. Merk, Daniel, et al. "De novo design of bioactive small molecules by artificial intelligence." *Molecular informatics* 37.1-2 (2018): 1700153.
7. Grisoni, Francesca, et al. "Combining generative artificial intelligence and on-chip synthesis for de novo drug design." *Science advances* 7.24 (2021): eabg3338.
8. Arús-Pous, Josep, et al. "Randomized SMILES strings improve the quality of molecular generative models." *Journal of cheminformatics* 11.1 (2019): 1-13.
9. Zheng, Shuangjia, et al. "QBMG: quasi-biogenic molecule generator with deep recurrent neural network." *Journal of cheminformatics* 11.1 (2019): 1-12.

The changes are pictured below in red (page 1 column 2):

Language models have been widely applied [21] with researchers using them for ligand based de novo design [22]. A few recent uses of language models include : targeting natural-product-inspired retinoid X receptor modulators [23], designing liver X receptor agonists [24], generating hit-like molecules from gene expression signatures [25], designing drug analogs from fragments [26], composing virtual quasi-biogenic compound libraries [27] and many others. Additional studies have highlighted the ability of language models in the low-data regime [28, 29] with improved performance using data augmentation [30].

Initially the brittleness of the SMILES string representation meant a single character could lead to an invalid molecules. This problem has been largely solved with more robust molecular string representations [31–34]. Additionally, with improved training methods, deep generative models based on RNNs consistently generate a high proportion of valid molecules using SMILES [6, 9, 35]. One area that has not been studied is the ability of language models and generative models to generate larger more complex molecules, or generate from chemical spaces with large ranges in size and structure. This is beneficial because of increased interest in larger more complex molecules for therapeutics [36].

In order to test the ability of language models, we formulate a series of challenging generative modeling tasks by constructing training sets of more complex molecules that exist in standard datasets [3, 35, 37]. In particular, We focus on the ability of language models to learn the distributional properties of the target datasets. We train language models on all tasks and include two baselines: CGAVE and JTVAE. The results demonstrate that language models are powerful generative models and can learn complex molecular distributions.

- Language models have been applied to design small drug-like molecules based on SMILES and peptides based on single-letter amino acid code. The present study now extends to the design of larger molecules (including peptides) based on the SMILES and SELFIES representation. This is a strong and novel aspect of this study and it would add value to follow up with further studies. Some suggestions that - in the opinion of this reviewer - would make interesting analyses:

Is a SMILES- or SELFIES-trained RNN able to conserve the peptide backbone, i.e., can the model design natural peptides? Does the model conserve natural amino acids?

Can a SMILES based RNN be trained to specifically design cyclic or bi-/tri-/...cyclic peptides?

Can such models design peptide mimetics? Such analysis would also benefit from direct comparisons between templates and designs, and an evaluation of training strategies.

Response 1.3: We thank the reviewer for suggesting these analyses. In response, we add some investigations on the topic of peptides and cyclic peptides – based on our results, we find that language models have a lot of potential for the task but would benefit from a more specific dataset for the target designed by an expert. The details and results are shown below (Figure 6b-d & page 7 columns 1-2):

TABLE II. Wasserstein distance metrics for LogP, SA, QED, MW, BT and NP between molecules from the training data and generated by the models for all three tasks. TRAIN is an oracle baseline- values closer to it are better.

1 and substructures, in FIG. 6a the RNN models adequately learn the distribution of substructures arising in the training molecules. Specifically the distribution for the number of: fragments, single and double atom fragments as well as single and multi-rings fragments in each molecule. As the training molecules get larger and occur less, both RNN models still learn to generate these molecules (FIG. 5a when molecular weight > 3000).

2 The dataset in this task contains a number of peptides and cyclic peptides that arise in PubChem, we visually analyze the samples from the RNNs to see if they are capable of preserving backbone chain structure and natural amino acids. We find that the RNNs often sample snippets of backbone chains which are usually disjoint broken up with other atoms, bonds and structures. In addition, usually these chains have standard side chains from the main amino acid residues but other atypical side chains do arise. In Figure 6b we show two examples of peptides with around 20 residues that are generated by the SM-RNN and SF-RNN and preserve backbone and amino acid structure. While there are many examples where both models do not preserve backbone and fantasize weird side-chains, it is very likely, that if trained entirely on relevant peptides the model could be used for peptide design. Even further, since these language models are not restricted to generating amino acid sequences the could be used to design any biochemical structure that mimic the structure of peptides or even replicate their biological behaviour. This makes them very applicable to design modified peptides [50], other peptide mimetics and complex natural products [51, 52]. The only requirement would be for a domain expert to construct a training dataset for specific targets. We conduct an additional study on how well the RNNs learned the biomolecular structures in the training data, in Figure 6d we see both RNNs match the distribution of essential

37 amino acid (AA) residues (which we find using a sub-
38 structure search). Lastly, it is also likely that the RNNs
39 could also be used to design cyclic peptides. To high-
40 light the promise of language models for this task we dis-
41 play some examples of molecules generated by the RNNs
42 that have the largest chemical similarity with colistin and
43 vancomycin (Figure 6c). The results in this task demon-
44 strate that language models could be used for designing
45 more complex biomolecules, but there is necessary addi-
46 tional work to be done for training strategies as well as
47 how to handle templates and designs.

D. Metrics

We also evaluate models on standard metrics in the literature: validity, uniqueness and novelty. Using the same 10K molecules generated from each model for each task we compute the following statistics defined in [16] and store them in TABLE. III: 1) validity: the ratio between the number of valid and generated molecules, 2) uniqueness: the ratio between the number of unique molecules (that are not duplicates) and valid molecules, 3) novelty: the ratio between unique molecules that are not in the training data and the total number of unique molecules. In the first two tasks (TABLE. III), JTVAE and CGVAE have better metrics with very high validity, uniqueness and novelty (all close to 1), here the SMILES and SELFIES RNN perform worse but the SELFIES RNN is close to their performance. The SMILES RNN has the worse metrics due to its poor grammar but is not substantially worse than the other models.

II. DISCUSSION

In this work, in effort to test the ability of chemical language models, we introduce three complex modeling tasks for deep generative models of molecules. Language models and graph baselines perform each task, which entails learning to generate molecules from a challenging dataset to learn. The results demonstrate that language models are very powerful, flexible models that can learn

FIG. 6. Large Scale Task **a** Histograms of fragment #, single atom fragment #, double atom fragment #, single ring fragment #, multi-ring fragment # (all per molecule) from molecules generated by the RNN models or from the training data **b** A peptide generated by the SM-RNN - SLFHKKLAVIGAVLKVLTTGLIA (left) and one generated by the SF-RNN - ERFRAQLGDEGSKEFVEEA (right) **c** Cyclic peptide-like molecules generated by the SF-RNN and SM-RNN that are closest in Tanimoto similarity to colistin (left) vancomycin (right). Both are highlighted in grey beside SF-RNN example(s) to the right and SM-RNN to the left **d** Histograms of amino acid (AA) # in largest found backbone structure in each molecule generated by the RNNs or training data.

Reviewer #2 (Remarks to the Author):

The manuscript by Flam-Shepherd et al. argues that generative models of molecules based on textual representations (aka language models) outperform generative models based on graphical representations (aka graph generative models). Their argument is based on a benchmark of two language models (both LSTMs, trained either on SELFIES or SMILES representations) against two graph generative models: constrained graph VAEs (Liu et al., NeurIPS 2018) and junction tree VAE (Jin et al., ICML 2018). The authors introduce three new tasks on which to benchmark these four models: specifically, generating molecules with high logP values, multi-modal property distributions, or with >100 heavy atoms. They demonstrate that the two language models are better at matching the target distributions on the first two tasks than either graph-based model.

I don't think the authors' results are necessarily novel in the strictest sense, because a handful of benchmarks have previously identified that language models outperform graph-based models (e.g., Mahmood et al., Nat. Commun. 2021). However, this has never really been a major focus of these analyses and I think it is potentially of enormous benefit for the field to highlight this more directly.

Response 2.1: Thank you we have added this reference, we agree that it is known now that language models have been shown to perform well and are comparable to other models including graph generative models. The main contribution of our work is to highlight some more complex molecular distributions (than standard datasets like ZINC, MOSES, ChEMBL) where language models excel and other widely used generative models have difficulties.

With that being said, the major flaw in the paper is the level of evidence provided for the broad claim that language models outperform graph-based models. The authors have really only shown that language models outperform the two specific graph-based models evaluated here. Dozens of graph-based models have been described, as helpfully reviewed by Mercado et al., Applied AI Lett. 2020, and it is not clear why these specific two were selected (they are claimed to be “state of the art” but no specific evidence supporting this claim is provided). While I don’t think it is either realistic or necessary to benchmark every single graph-based model that has ever been described, I don’t think that the findings presented here convincingly support the more general claim. In my view, that would require benchmarking several (at least a half dozen) of the other prominent graph-based models for which source code is publicly available - for example GraphINVENT, MolGAN, GraphVAE, MolecularRNN, MolMP/MolRNN, HierVAE, and NeVAE - in order to establish a more general trend.

Response 2.2: Thank you for the review article– we have cited it, we also considered the baselines you mention and a few additional ones. None of the additional baselines perform better than the language models. The results are shown below (on the next two pages in **Action 2**) and have been added in the revision on pages 13-14.

Action 1: We also revise the paper to have a more nuanced discussion on the topic of the ability of Language Models versus Graph Generative Models. We avoid broad claims and focus on a more specific discussion comparing the baselines and language models– for instance (page 9 column 1 paragraph 3):

We provide additional comparison with other graph generative models [8, 12, 17, 19, 54–59] in the supplementary and find that most do not scale to these generative tasks and for the ones that do– the language models perform better than those additional baselines as well. However, most graph generative models have the more difficult task of generating both the atom and bond information– while a language model only has to generate a single sequence. Given this– it is natural that language models displays such flexible capacity and the evaluated graph generative models do not. Outside of molecular design some graph generative models have attempted to scale to larger graphs [60, 61] but these models have not been augmented for molecules. The results here do highlight the fact that many widely used graph generative models are designed only for small drug-like [40] molecules and do not scale to larger more complex molecules. On the other hand, while language models can scale and flexibly generate larger molecules, graph generative models are more interpretable [54, 58] which is important for drug and material discovery.

Action 2: We add a discussion, some metric tables, and distribution plots for the additional baselines (pages 13-14)

Additional Baselines: We experiment with additional graph generative model baselines on all tasks. These include HierVAE [55], GCPN [11], GRAPH AF [68], GENRIC [12], CNF [54], Molecular RNN (MRNN) [56], GRAPHVAE [16], NAT-GRAPHVAE [69], MOLGAN [18], GRAPH NVP [58], DGMG [8], MOLMP [57], GRAPHINVENT [53]. From these models, most of the single shot generative models do not scale – from MOLGAN, GRAPHVAE, NAT-GRAPHVAE, GRAPH NVP. None of these models including GCPN were able to achieve better than 1% valid, unique and novel – meaning they are unable to generate molecules from the training distribution. Furthermore, all of the autoregressive graph generative models (DGMG, MolMP, GRAPH INVENT) were unable to handle the larger molecules – even in the LogP and Multi-distribution tasks. Training on these larger datasets exacerbate the stability issues [8] these models suffer from – making them unable to stably train to completion. The baselines that were able to train could only handle the LogP and multi-distribution tasks, these include: two discrete normalizing flow models CNF [54] and GRAPH AF [68], GENRIC which employs a Markov chain, MRNN or Molecular RNN which uses RNNs to generate atoms and bonds and HIERVAE which extends JTVAE to larger common motifs or substructures. All baselines have high scoring standard metrics (Table II) but their wasserstein distance metrics are much further from the Train Oracle than the RNNs (Table I). HIERVAE and MRNN stand out and are higher scoring than GENRIC, CNF and GRAPH AF – HIERVAE even beats the SF-RNN on SA and NP but not the SM-RNN. Indeed, from the distribution plot in Figure S1a for the LogP task we can see that MRNN and HIERVAE are closer to the training distribution than the additional baselines but nearly as close as the RNNs. For the multi-distribution task, the closest are MRNN and CNF, shown in the distribution plot in Figure S1c – where MRNN learns all of the modes (but poorly) while CNF entirely misses the CEP mode. In contrast the RNNs, perfectly learn all four modes (Figure S1b).

Task	Samples	LogP	SA	QED	MW	BCT	NP
LogP	TRAIN	0.020	0.0096	0.0029	1.620	7.828	0.013
	SM-RNN	0.095	0.0312	0.0068	3.314	21.12	0.054
	SF-RNN	0.177	0.2903	0.0095	6.260	25.00	0.209
	HIERVAE	0.661	0.0464	0.0710	51.73	141.9	0.079
	MRNN	0.769	1.2321	0.0710	58.27	142.9	0.898
	GRAPHAF	3.534	1.8820	0.2413	164.7	664.4	1.206
	CNF	2.773	3.4727	0.1879	37.87	174.7	1.456
	GENRIC	2.764	1.3626	0.1092	81.41	308.0	1.286
Multi	TRAIN	0.048	0.0158	0.0020	2.177	14.15	0.010
	SM-RNN	0.081	0.0246	0.0059	5.483	21.19	0.012
	SF-RNN	0.286	0.1791	0.0227	11.35	68.81	0.079
	HIERVAE	2.356	0.2151	0.1024	157.7	687.0	0.175
	MRNN	1.519	0.6644	0.0593	97.92	400.1	0.598
	GRAPHAF	3.140	1.9122	0.1174	106.1	971.7	0.723
	CNF	2.378	2.0793	0.0991	61.87	436.7	1.070
	GENRIC	1.623	2.0029	0.0827	105.7	445.3	0.787

TABLE I. Wasserstein distance metrics for LogP, SA, QED, MW, BT and NP between molecules from the training data and generated by the additional baselines and RNNs for all three tasks. TRAIN is an oracle baseline- values closer to it are better.

Task	Metric	SM-RNN	SF-RNN	HIERVAE	MRNN	GRAPHAF	CNF	GENRIC
LogP	validity	0.941	1.000	1.000	1.000	1.000	1.000	1.000
	uniqueness	0.987	1.000	1.000	0.999	0.906	1.000	0.886
	novelty	0.721	0.871	1.000	0.994	1.000	1.000	0.993
Multi	valid	0.969	1.000	1.000	0.999	1.000	1.000	0.997
	uniqueness	0.996	0.989	0.938	0.999	0.985	1.000	0.912
	novelty	0.937	0.950	1.000	1.000	1.000	1.000	0.998

TABLE II. Standard Metrics of validity, uniqueness and novelty of molecules generated by all models in every task. Closer to 1.0 indicates better performance.

FIG. S1. **Additional Baselines** **a** For the LogP task, the histogram and KDE of penalized logP of training molecules along with KDEs of molecular weight of molecules generated from additional baselines model that could generate samples. **b** The Histogram and KDE of molecular weight of training molecules along with KDEs of molecular weight of molecules generated from the training data and RNNs. **c** The Histogram and KDE of molecular weight of training molecules along with KDEs of molecular weight of molecules generated from the training data and from additional baselines.

A second concern is the authors' argument that their own SELFIES representation "seems to improve the performance of language models in every task," as compared to a language model trained on SMILES. However, on 10 of 10 distribution learning tasks presented in Tables I and II, SM-RNN achieves better performance than SF-RNN. Thus, this statement is not supported and in fact is contradicted by the data presented in the manuscript.

Response 2.4: We have revised this paragraph to give a more clear and nuanced evaluation of the results:

Comparison of SELFIES and SMILES. Both the SM-RNN and SF-RNN perform well in all tasks, better than the baselines. We notice that the SF-RNN has better standard metrics (Table III) in every task, but the SM-RNN has better Wasserstein distance metrics (Table II). Furthermore, the SF-RNN has better novelty than the SM-RNN—this may mean that the SELFIES grammar leads to less memorization of the training dataset in chemical language models. This could also help explain why the SF-RNN has better standard metrics but worse Wasserstein metrics than the SM-RNN. In addition, data augmentation and random SMILES [30] could be used to improve the novelty score of the SM-RNN. In future, it would be valuable to have a more comprehensive evaluation of the use of SMILES and SELFIES representations in deep generative models.

A final issue is the level of detail provided in the Methods section, which is insufficient to reproduce all the results of the paper - for example, how was random search executed? How were the parameter grids defined? How were the evaluation metrics calculated? Ideally, the authors would just provide the source code that was used to execute these searches and evaluate model performance.

Response 2.5: We have added additional methods details necessary (page 10 column 1):

Hyper-parameter Optimization. For hyper-parameter optimization we use the simplest most effective method—namely random search [62]. We randomly sample from discrete grids of hyper-parameters with equal probability of selection for each value. The values are roughly equally spaced with 3-5 values in each grid. The upper and lower bounds for each hyper-parameter are defined as such : learning rate $\in [0.001, 0.0001]$, hidden units $\in [100, 1000]$, layer number $\in [1, 5]$, dropout (probability) in $[0.0, 0.5]$. We don't optimize the number of epochs— we just use the default value for the baseline models used during training on other datasets (MOSES, ZINC or ChEMBL).

Model Selection Criteria. There are many model selection criteria possible, for example— the MOSES benchmark [35] suggest the Frechet Distance, however, this and other performance metrics have been shown to have issues [63]. We evaluate and select models using all metrics employed in combination with the distribution plots. First we compile the top 10% of models with highest validity, uniqueness and novelty. Then we plot distribution plots for the main property of interest (ie penalized logP for LogP task and molecular weight for others)— then take the model that has the closest distribution to the training distribution and scores the lowest on largest number of the six Wasserstein distance metrics.

Further Details: Language models are implemented in Python 3 with PyTorch [64] molecules are processed and relevant properties are computed using RDKit [38]. Wasserstein distances are computed using SciPy [65] as `scipy.stats.wasserstein.distance` based on [66] — also known as the earth mover's distance, it can be viewed as the minimum amount of distribution weight that must be moved, multiplied by the distance— in order to transform samples from one distribution into samples from the another.

Minor points:

1. The statement in the abstract that graph generative models “typically achieve state of the art results” is not, to my knowledge, true, and in fact would seem to be contradicted by the results in this manuscript. It would be nice if the authors could clarify what data supports this claim or else remove it.

We have removed this sentence.

2. Much is made in the introduction of the notion that generation of invalid SMILES makes it difficult to train and apply language models. I can't say that I understand why this is a major obstacle, as it would appear straightforward to filter out the invalid SMILES strings. I wonder if the authors can articulate some scenarios where it is essential that all generated molecules be valid (i.e. where filtering out invalid SMILES is not possible).

Response 2.6: We agree, this is over-emphasized – we have made some changes to the introduction focusing much less emphasis on this, which is not the focus of this work (page 9 column 1 paragraph 1):

Initially the brittleness of the SMILES string representation meant a single character could lead to an invalid molecules. This problem has been largely solved with more robust molecular string representations [31–34]. Additionally, with improved training methods, deep generative models based on RNNs consistently generate a high proportion of valid molecules using SMILES [6, 9, 35]. One area that has not been studied is the ability of language models and generative models to generate larger more complex molecules, or generate from chemical spaces with large ranges in size and structure. This is beneficial because of increased interest in larger more complex molecules for therapeutics [36].

3. The authors assert that the three new benchmark tasks introduced here are “especially challenging.” I can see why modelling very large molecules could be challenging, but I'm not convinced that modelling especially lipophilic or chemically diverse molecules should present a particular challenge.

Response 2.7: Even the LogP and Multi-distribution datasets are fairly larger than standard datasets like ZINC and MOSES with more atoms and rings. They also have a larger range between the smallest molecule and the largest molecules. We have added another table highlighting this (page 2 column 1 & shown below). We also remove the word “especially” to avoid exaggeration.

	# Atoms			# Rings		
	Min	Mean	Max	Min	Mean	Max
Zinc	6	23.2	38	0	2.8	9
Moses	8	21.6	27	0	2.6	8
Log p	12	34.7	78	0	4.2	37
Multi	7	31.1	106	0	5.3	23
Large	101	140.1	891	0	11.2	399

TABLE I. **Dataset Statistics** for all three tasks.

In Table I there are some summary statistics of atom and ring number in all datasets compared with two standard datasets Zinc [3] and Moses [35]. All tasks involve larger molecules with more substructures and contain a larger range of atom and ring number per molecule.

4. A table of Wasserstein metrics for the large scale task is missing.

Response 2.8: Thank you– we have consolidated all Wasserstein metrics into one table, including the metrics from the large scale task (Table II on page 8 column 1):

Task	Samples	LogP	SA	QED	MW	BCT	NP
LogP	TRAIN	0.020	0.0096	0.0029	1.620	7.828	0.013
	SM-RNN	0.095	0.0312	0.0068	3.314	21.12	0.054
	SF-RNN	0.177	0.2903	0.0095	6.260	25.00	0.209
	JTVAE	0.536	0.2886	0.0811	35.93	76.81	0.164
	CGVAE	1.000	2.1201	0.1147	69.26	141.2	1.965
Multi	TRAIN	0.048	0.0158	0.0020	2.177	14.149	0.010
	SM-RNN	0.081	0.0246	0.0059	5.483	21.118	0.012
	SF-RNN	0.286	0.1791	0.0227	11.35	68.809	0.079
	JTVAE	0.495	0.2737	0.0343	27.71	171.87	0.109
	CGVAE	1.617	1.8019	0.0764	30.31	183.58	1.376
Large	TRAIN	0.293	0.030	0.0003	18.92	85.04	0.005
	SM-RNN	1.367	0.213	0.0034	124.49	363.0	0.035
	SF-RNN	1.095	0.342	0.0099	67.322	457.5	0.111
	JTVAE	-	-	-	-	-	-
	CGVAE	-	-	-	-	-	-

TABLE II. Wasserstein distance metrics for LogP, SA, QED, MW, BT and NP between molecules from the training data and generated by the models for all three tasks. TRAIN is an oracle baseline- values closer to it are better.

5. In the Discussion, the authors suggest their results raise the possibility that language models are overfit to the training data. It was not clear to me exactly what aspects of their results suggested this. If overfitting is a concern, the framework based on the GDB-13 database presented by Arus-Pous et al., J. Cheminform. 2019 could be useful to test the generalization capacity of these models.

Response 2.9: Our concern was the lower novelty score for the RNNs, particularly the SM-RNN. However, they are only slightly lower and so we have revised this claim and cited your suggested article as a possible way to improve performance (page 9 column 1 paragraph 1):

II). Furthermore, the SF-RNN has better novelty than the SM-RNN– this may mean that the SELFIES grammar leads to less memorization of the training dataset in chemical language models. This could also help explain why the SF-RNN has better standard metrics but worse Wasserstein metrics than the SM-RNN. In addition, data augmentation and random SMILES [30] could be used to improve the novelty score of the SM-RNN. In future, it would be valuable to have a more comprehensive evaluation of the use of SMILES and SELFIES representations in deep generative models.

REVIEWERS' COMMENTS

Reviewer #1 (Remarks to the Author):

The authors have addressed my suggestions and the manuscript appears suitable for publication.

Reviewer #2 (Remarks to the Author):

The authors have done an excellent job responding to the reviewer comments. I very much appreciate the extensive benchmarking of other graph-based generative models they have added to the revised manuscript. This new data substantially strengthens their argument that language models outperform graph generative models. I think this will be a very impactful paper in the field and recommend its publication in Nature Communications.

I have only one more suggestion, which is that the authors could consider changing the title to more clearly reflect what I would consider to be the main result of their manuscript: namely, that language models are not just able to learn complex molecular distributions, but that they outperform graph generative models on this task. I think this would also be worth highlighting in the abstract and introduction. In particular, I am not sure that describing graph-based generative models as “baselines” does full justice to the authors’ findings, as these models are widely seen - correctly or not - to be the most advanced methods that are currently available.

REVIEWERS' COMMENTS

Reviewer #1 (Remarks to the Author):

The authors have addressed my suggestions and the manuscript appears suitable for publication.

Reviewer #2 (Remarks to the Author):

The authors have done an excellent job responding to the reviewer comments. I very much appreciate the extensive benchmarking of other graph-based generative models they have added to the revised manuscript. This new data substantially strengthens their argument that language models outperform graph generative models. I think this will be a very impactful paper in the field and recommend its publication in Nature Communications.

I have only one more suggestion, which is that the authors could consider changing the title to more clearly reflect what I would consider to be the main result of their manuscript: namely, that language models are not just able to learn complex molecular distributions, but that they outperform graph generative models on this task. I think this would also be worth highlighting in the abstract and introduction. In particular, I am not sure that describing graph-based generative models as “baselines” does full justice to the authors’ findings, as these models are widely seen - correctly or not - to be the most advanced methods that are currently available.

We thank the reviewers for their comments. In response to the suggestions of reviewer #2— we have added more emphasis that language models outperform graph generative models— in the introduction, abstract, and discussion sections.